# Pregnancy in a Non-Communicating Rudimentary Horn of Unicornuate Uterus

**DOI:** 10.3390/diagnostics12030759

**Published:** 2022-03-21

**Authors:** Yi-Cih Ma, Kim-Seng Law

**Affiliations:** 1Department of Obstetrics and Gynecology, Tung’s Taichung MetroHarbor Hospital, Taichung 435, Taiwan; yicihmama@gmail.com; 2Department of Nursing, Jenteh Junior College of Medicine, Nursing and Management, Miaoli 356, Taiwan; 3Department of Post-Baccalaureate Medicine, College of Medicine, National Chung Hsing University, Taichung 402, Taiwan

**Keywords:** non-communicating, pregnancy, rudimentary horn, unicornuate

## Abstract

We report a rare case of non-communicating rudimentary horn pregnancy (RHP). The patient presented with lower abdominal pain and underwent laparoscopic surgery in which the gestational tissue was removed without excision of the rudimentary horn and ipsilateral fallopian tube. Unicornuate uteri often coexist with rudimentary horns, most of which are non-communicating. RHP is rare, and symptomatic women tend to complain of abdominal pain. Once RHP is suspected, the clinician should monitor the patient for signs of hypovolemic shock, such as hypotension, because the RHP can rupture owing to the poorly developed musculature. Early surgical intervention with removal of the rudimentary horn along with the ipsilateral fallopian tube is generally suggested to prevent future ectopic pregnancy. The theory of sperm transmigration from the contralateral oviduct has been hypothesized in non-communicating RHP.

## 1. Introduction

The incidence of unicornuate uterus is 0.4%, with approximately 84% of the women having a contralateral rudimentary horn [1]. In one review, 92% of 366 cases of unicornuate uterus with rudimentary horn were reported to be non-communicating, and 36% were associated with renal anomalies [2]. Rudimentary horn pregnancy (RHP) is even rarer, with a reported incidence ranging from 1 in 76,000 to 1 in 150,000 [1]. Approximately 40% of women with RHP are asymptomatic, and abdominal pain is the most common clinical symptom [3]. Early surgical intervention with removal of the rudimentary horn containing the gestational sac along with the ipsilateral fallopian tube has been recommended for cases of rudimentary horn pregnancy [2]. Laparoscopic surgery has been successfully used in such situations [4]. We report a rare case of a non-communicating right RHP owing to transmigration of the sperm from the contralateral oviduct; the gestational product was extracted completely, and recovery was uneventful.

## 2. Case Report

A 37-year-old woman (gravida 2, para 1) presented with sudden onset lower abdominal cramping pain associated with cold sweat and diarrhea. She gave birth to her first baby through caesarian section owing to breech presentation at a gestational age of 38 5/7 weeks. According to the patient, no structural Müllerian abnormality was recalled at that time. Fifteen days before admission, the patient noted that she had missed her expected menstrual period. Pregnancy was confirmed by the doctor. Transabdominal sonography did not reveal any intrauterine gestational sac at the time, and neither was an intrauterine sac found ten days later, with a serum β-HCG level of 8487 mIU/mL. Under the suspicion of ectopic pregnancy, she was referred to our outpatient department at a gestational age of 7 1/7 weeks in accordance with her last menstrual period. Transabdominal sonography revealed a right adnexal mass adjacent to the uterus, with a size of 2 cm and a hypoechoic sac-like structure (Figure 1). Her serum β-HCG level was elevated to 20,124 mIU/mL 72 h after the first serum test. The initial clinical impression was a right adnexal ectopic pregnancy. Surgical intervention was suggested, but the patient opted for methotrexate treatment, despite being advised as to the high possibility of failure. The patient provided informed consent for publication of the data.

A single intramuscular injection of MTX (50 mg) was administered. The patient returned two days later because of another episode of lower abdominal cramping pain. Transabdominal sonography revealed scant fluid collection over the cul-de-sac. Her serum β-HCG level was 27,472 mIU/mL. The patient was admitted for laparoscopic intervention. Whole abdominal computed tomography (CT) without contrast was performed before surgery, which showed a right adnexal mass in compliance with ectopic pregnancy (Figure 2).

Upon entering the pelvic cavity, a unicornuate uterus with a right rudimentary horn was identified, with the attachment of round ligaments as well as ovarian ligaments and the fallopian tube to the horn (Figure 3). There was no hemoperitoneum, except for straw-like fluid collection over the cul-de-sac. The rudimentary horn was connected to the main uterus by a thick and seemingly obliterated fibrous band (Figure 3). A horizontal incision over the rudimentary horn was made with unipolar scissors, which exposed chorionic villi-like tissue. The presumed gestational tissue was completely removed, and the muscular layer was closed with 1-O Vicryl (Figure 3) after hemostasis was performed. In this case, neither the right rudimentary horn nor the fallopian tube was excised because of the uncertainty of the diagnosis.

The patient was discharged uneventfully, and her serial serum β-HCG levels were followed up until normal. The final pathology report indicated gestational products. An office hysteroscopy was performed after the commencement of her first menses, with no opening found between the uterine cavity and the horn. A left unicornuate uterus with non-communicating right RHP, probably conceived through transmigration of the sperm via the left oviduct, was the final diagnosis.

## 3. Discussion

Mullerian abnormalities are reported in 0.17% of fertile women and 3.5% of infertile women, and the unicornuate uterus is found in 0.4% of all women [1]. According to the American Fertility Society, there are four variants of unicornuate uterus classified based on the presence or absence of the horn, whether it is a communicating or non-communicating horn, and whether it is non-functional. Approximately 84% of unicornuate uteri have a contralateral rudimentary horn [1]. A review of 366 cases of unicornuate uteri with rudimentary horn, from 1966 to 2003, showed functional endometrium in 82% of these cases [2]. Non-communicating functional horns tend to cause dysmenorrhea at menarche by forming hematometra or endometriosis [2].

The other female genital tract anomaly classification is the ESHRE/ESGE system based on the main uterine findings (U0–U6) and cervical/vaginal anomalies (C0–C4; V0–V4); the present case was categorized as U4 C0 V0 [3].

The overall sensitivity of ultrasonography for a definite diagnosis of rudimentary horn was 26%, and only 14% (95% CI, 7–23%) could be diagnosed before symptom presentation [2]. Magnetic resonance imaging, three-dimensional ultrasonography, hysterosalpingography, and hysteroscopy have been advocated to aid in the accurate diagnosis of a unicornuate uterus with a rudimentary horn.

RHP is rare, with a reported incidence ranging from 1 in 76,000 to 1 in 150,000 [1]. Approximately 40% of women with RHP are asymptomatic, whereas symptomatic women tend to complain of abdominal pain [4]. Massive bleeding with shock/unstable vital signs and even death may occur following RHP rupture due to the poorly developed musculature. Moreover, a higher rate of placenta accreta has also been reported [2], which may cause postpartum hemorrhage. In view of the possible maternal morbidity and mortality, immediate surgical intervention after the diagnosis of early RHP has been recommended. Complete removal of the rudimentary horn along with the ipsilateral fallopian tube is generally recommended, although we removed the intracavitary tissue because of uncertainty in diagnosis. The placement of a running suture and electrocauterization of the cavity for hemostasis following retrieval of the contents would theoretically prevent a second rudimentary pregnancy; however, a right tubal pregnancy in the future cannot be completely ruled out [2]. Laparoscopic surgery has been successfully used in such situations [4]. The use of preoperative methotrexate to terminate early pregnancy before laparoscopic intervention has been reported in the literature [5,6,7].

The theory of transmigration of sperm from the contralateral oviduct has been advocated in an article by Ansari et al. [8], with confirmation by corpus luteum on the ipsilateral adnexae. This proposition can be verified by our finding of the absence of a “tunnel” connecting the rudimentary horn with the unicornuate cavity. Close follow up of the following pregnancy in this particular case should be performed because of retention of the rudimentary horn and fallopian tube.

## Figures and Tables

**Figure 1 diagnostics-12-00759-f001:**
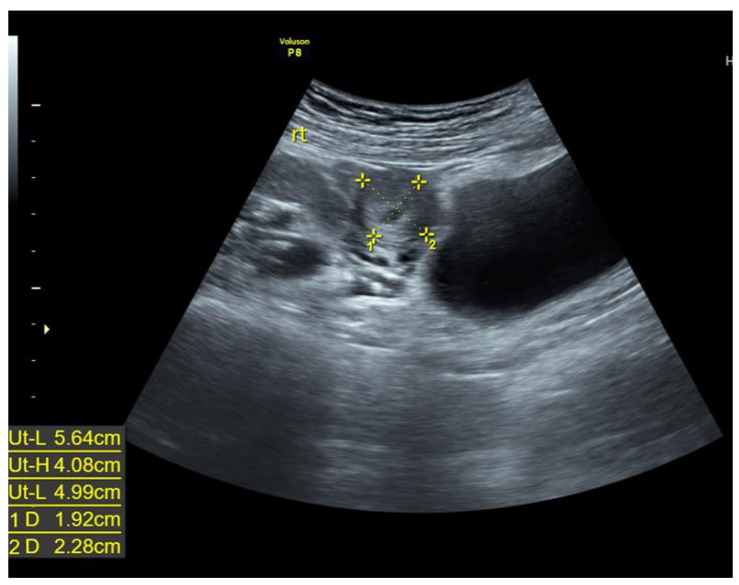
A 2-cm right adnexal mass adjacent to the uterus with hypoechoic sac-like structure under trans-abdominal sonography.

**Figure 2 diagnostics-12-00759-f002:**
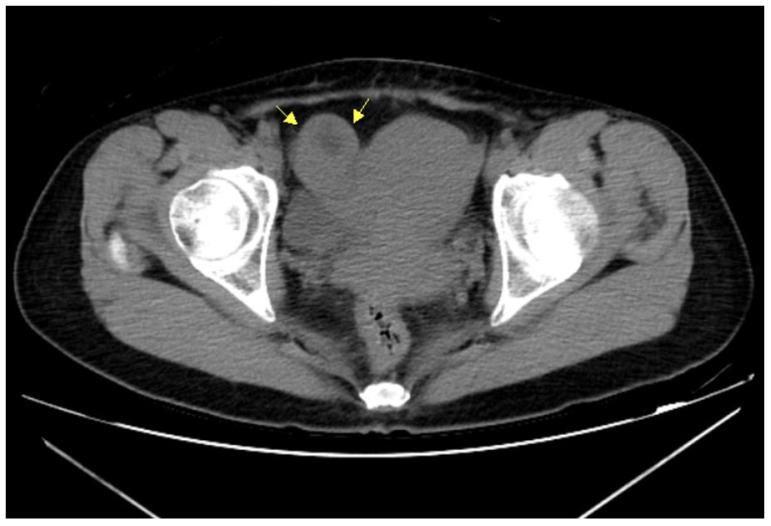
Non-contrast pelvic computed tomography also revealed a right adnexal mass near the uterus (arrows).

**Figure 3 diagnostics-12-00759-f003:**
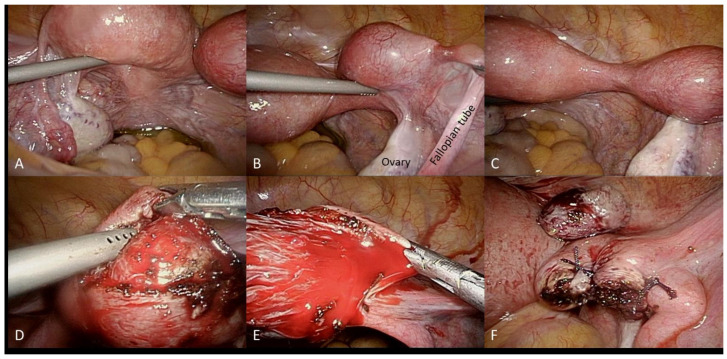
Under laparoscopic operation, the left unicornuate uterus (**A**) and right rudimentary horn (**B**) were identified through the attachment of corresponding ipsilateral fallopian tube, and ovarian and round ligaments. The unicornuate uterus and rudimentary horn were connected by a broad, dense fibrous band (**C**). The gestational tissue was completely removed from the rudimentary horn by a transmuscular horizontal incision (**D**,**E**). The muscular layer was then closed using 1-O Vicryl suture with the excised gestational tissue placed above (**F**).

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
