# Peer review of "Pregnancy in a Non-Communicating Rudimentary Horn of Unicornuate Uterus"

_diagnostics, 2022, doi:10.3390/diagnostics12030759_

Round 1

Reviewer 1 Report

The main aim of this case report is to draw attention over the probability of abnormal pregnancy associated with uterine malformation and its prognosis. However, another purpose would be the correct management of such cases. Therefore, i would like to ask the authors some questions:

  1. You mentioned that you only used transabdominal ultrasound. Why didn’t you perform transvaginal ultrasonography as it is more accurate in adnexal masses diagnosis?
  2. Why did you approach this surgical management? Were there any problems with lack of consent due to the misdiagnosis at the beginning? Did you have a suspicion of a diagnosis of uterine malformation before proceeding with the surgery? The gold-standard, as cited in many studies, in such cases is removing the rudimentary horn along with the ipsilateral salpingectomy. Please elaborate on why you chose to proceed as you described, because not knowing the diagnosis is not a justifying reason. In my point of view there is no reason to keep the rudimentary horn, as its existence can only be burdened by several complications such as subsequent pregnancies with spontaneous rupture, haemoperitoneum, hematometra, endometriosis etc.
  3. The ESHRE classification of female genital tract congenital anomalies should be mentioned. I strongly suggest you classify the discovered malformation according to ESHRE classification.
  4. Uterine malformations may be associated with renal malformations. In particular, the unicornuate uterus is usually associated with renal anomalies, contralateral renal agenesis being the most common anomaly. Horseshoe kidney, unilateral medullary sponge kidney, and double renal pelvis are infrequent.In addition to renal malformations, uterine abnormalities can also be associated with other types of malformations, see VACTERL association which comprises more birth defects that co-occur. Have you investigated the patient for more than one type of malformation, especially renal malformations? The co-existence of malformations should be mentioned.

Author Response

We greatly appreciate the comments by the two reviewers on our article, titled “Pregnancy in a Non-communicating Rudimentary Horn of Unicornuate Uterus“. Please find below our point-by-point responses to the same.

1. You mentioned that you only used transabdominal ultrasound. Why didn’t you perform transvaginal ultrasonography as it is more accurate in adnexal masses diagnosis?

Response:

Transabdominal ultrasound was used in our case because the patient presented with sudden onset lower abdominal pain initially at the ER and on the second visit. Our out-patient clinic has a large working population, and the ultrasound was performed by the visiting doctor personally at the bedside; the second ultrasound revealed a right adnexal mass with sac-like structures. However, we agree with your point that the transvaginal route is a better way to elucidate an adnexal mass.

2. Why did you approach this surgical management? Were there any problems with lack of consent due to the misdiagnosis at the beginning? Did you have a suspicion of a diagnosis of uterine malformation before proceeding with the surgery? The gold-standard, as cited in many studies, in such cases is removing the rudimentary horn along with the ipsilateral salpingectomy. Please elaborate on why you chose to proceed as you described, because not knowing the diagnosis is not a justifying reason. In my point of view there is no reason to keep the rudimentary horn, as its existence can only be burdened by several complications such as subsequent pregnancies with spontaneous rupture, haemoperitoneum, hematometra, endometriosis etc.

Response:

We approached this surgical management based on the suspicion of a right adnexal ectopic pregnancy, and there was no problem with lack of consent in this scenario. We did not expect an anomaly beforehand, and due to the unexpected finding, a more conservative approach was taken instead of extirpation of the whole rudimentary horn as suggested in most articles. With the running suture performed after extraction of the intracavitary content followed by heavy bleeding, adhesion would probably prevent subsequent pregnancy inside the horn; however, there is still a slight risk of right tubal pregnancy, and we have mentioned this in our discussion section as follows : “We removed the intracavitary tissue because of uncertainty in diagnosis. The placement of a running suture and electrocauterization of the cavity for hemostasis following retrieval of the contents would theoretically prevent a second rudimentary pregnancy; however, a right tubal pregnancy in the future cannot be completely ruled out.”

3. The ESHRE classification of female genital tract congenital anomalies should be mentioned. I strongly suggest you classify the discovered malformation according to ESHRE classification.

Response:

The anomaly in our case, according to the ESHRE/ESGE classification, is U4 C0, corresponding to a hemi-uterus with a rudimentary non-communicating horn and a normal cervix. We have added the following sentence to the Discussion section of our manuscript: “The other female genital tract anomaly classification is the ESHRE/ESGE system based on the main uterine findings (U0–U6) and cervical/vaginal anomalies (C0–C4; V0–V4); the present case was categorized as U4 C0 V0.”

4. Uterine malformations may be associated with renal malformations. In particular, the unicornuate uterus is usually associated with renal anomalies, contralateral renal agenesis being the most common anomaly. Horseshoe kidney, unilateral medullary sponge kidney, and double renal pelvis are infrequent.In addition to renal malformations, uterine abnormalities can also be associated with other types of malformations, see VACTERL association which comprises more birth defects that co-occur. Have you investigated the patient for more than one type of malformation, especially renal malformations? The co-existence of malformations should be mentioned.

Response:

We investigated the associated anomaly in our patient, who was found to have no other anomalies in either the urinary system or other constitutional organs.

Reviewer 2 Report

Your paper reports on a rare obstetric complication. The pathology per se in interesting, but I have some remarks.

  1. language: please check your tenses throughout the whole manuscript, you switch, resulting in logical errors. Obvious mistakes: '...neither was uterine sac found...', '...of suspicious of...', '...according to the presence or absent...'.
  2. Please explain: ' ...we had explained the high possibility of failure...'? The opposite is true, MTX has a very high succes rate in ectopic pregnancy.
  3. The connection does visually not seem obliterared, did you check the patency during laparoscopy?
  4. You did not resect due to 'uncertainty of diagnosis'. To enhance the originality of your report I would recommend that you discuss that subject and give the state-of-art as well as pros and cons in case of doubt.

Author Response

We greatly appreciate the comments by the two reviewers on our article, titled “Pregnancy in a Non-communicating Rudimentary Horn of Unicornuate Uterus“. Please find below our point-by-point responses to the same.

1. language Please check your tenses throughout the entire manuscript. You switch, resulting in logical errors. Obvious mistakes: '...neither was the uterine sac found..., '...of suspicious of...', “” ”'...according to the presence or absent...'.

Response:

We will send this article once again to another English editing company for grammatical corrections.

2. Please explain: ' ...we had explained the high possibility of failure...'? The opposite is true, MTX has a very high succes rate in ectopic pregnancy.

Response:

The failure rate for a single-dose MTX in an initially high serum bhcg is rather high, as has been confirmed by several studies, as cited below.

Factors that impact efficacy

  • High hCG concentration – High serum hCG concentration is the most important factor associated with MTX treatment failure (Table 1). Patients with a high baseline hCG concentration (>5000 mIU/mL) are more likely to require multiple courses of MTX therapy or experience treatment failure.

A systematic review of observational studies included 503 patients, and the outcomes of single-dose MTX therapy were stratified according to the initial hCG concentration. There was a statistically significant increase in failure rates in patients with initial hCG levels of >5000 milli-international units/mL compared with those who had initial levels of less than 5000 milli-international units/mL (odds ratio [OR] 5.5, 95% CI 3.0-9.8).

3. The connection does visually not seem obliterared, did you check the patency during laparoscopy?

Response:

The dense fibrous connection between the uterus and rudimentary horn, as seen under laparoscopy, was seemingly obliterated and confirmed postoperatively with both hysteroscopy and HSG. We did not check for patency during the surgery.

4. You did not resect due to the uncertainty of the diagnosis. To enhance the originality of your report, I would recommend that you discuss that subject and give the state-of-art as well as the pros and cons in case of doubt.

Response:

We will elaborate on this issue in our discussion section as follows and thank you again for the suggestion.” Instead, we removed the intracavitary tissue owing to uncertainty in the diagnosis .With the running suture placed and electrocauterization over the cavity for hemostasis following the retrieval of the content, it was theoretically having little chance of a second rudimentary pregnancy, although a right tubal pregnancy could not be totally ruled out in the future.”

Round 2

Reviewer 2 Report

Your review was extensive and improved the quality of your paper, well done.